# Latent Thought Models with Variational Bayes Inference-Time Computation

**Deqian Kong** [1 2 †]   **Minglu Zhao** [1 †]   **Dehong Xu** [1 †]   **Bo Pang** [3]   **Shu Wang** [1]   **Edouardo Honig** [1]   **Zhangzhang Si** [4]
**Chuan Li** [2]   **Jianwen Xie** [2 ‡]   **Sirui Xie** [1 ‡]   **Ying Nian Wu** [1 ‡]

## Abstract

We propose a novel class of language models, Latent Thought Models (LTMs), which incorporate explicit latent thought vectors that follow an explicit prior model in latent space. These latent thought vectors guide the autoregressive generation of ground tokens through a Transformer decoder. Training employs a dual-rate optimization process within the classical variational Bayes framework: fast learning of local variational parameters for the posterior distribution of latent vectors (inference-time computation), and slow learning of global decoder parameters. Empirical studies reveal that LTMs possess additional scaling dimensions beyond traditional Large Language Models (LLMs), such as the number of iterations in inference-time computation and number of latent thought vectors. Higher sample efficiency can be achieved by increasing training compute per token, with further gains possible by trading model size for more inference steps. Designed based on these scaling properties, LTMs demonstrate superior sample and parameter efficiency compared to autoregressive models and discrete diffusion models. They significantly outperform these counterparts in validation perplexity and zero-shot language modeling tasks. Additionally, LTMs exhibit emergent few-shot in-context reasoning capabilities that scale with model size, and achieve competitive performance in conditional and unconditional text generation. The project page is available at https://deqiankong.github.io/blogs/ltm.

†Equal contribution ‡Equal advising [1]UCLA [2]Lambda, Inc. [3]Salesforce Research [4]KUNGFU.AI. This work was partially conducted while D. K. was an intern at Lambda, Inc. Correspondence to: Deqian Kong <deqiankong@ucla.edu>.

*Proceedings of the 42$^{nd}$ International Conference on Machine Learning*, Vancouver, Canada. PMLR 267, 2025. Copyright 2025 by the author(s).

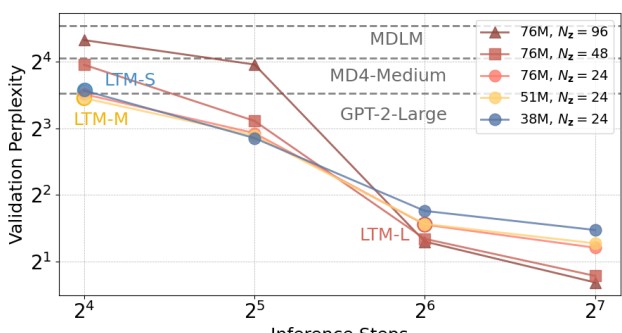

Figure 1: **Analysis of model scaling behavior** of validation perplexity across model size, inference steps, and the number of latent thought vectors $N_{\mathbf{z}}$. Autoregressive and diffusion baselines are plotted as dashed lines.

## 1. Introduction

Recent years have witnessed remarkable advancements in the field of natural language processing, primarily driven by the development of large language models (LLMs). These models, exemplified by GPT-3 (Brown et al., 2020), PaLM (Chowdhery et al., 2022), and their successors, have demonstrated impressive capabilities across a wide range of language tasks, from text generation and translation to question answering and complex reasoning. Their performance has often approached, and in some cases even surpassed, human-level competence in specific domains.

The remarkable success of LLMs is underpinned by well-established scaling laws (Kaplan et al., 2020; Hoffmann et al., 2022), which predict performance improvements with increased model and data size. The induced equations reveal that larger models achieve significantly higher sample efficiency (evaluated by the number of training tokens for achieving certain performance), making it computationally optimal to train very large models and stop before convergence. However, as model sizes grow rapidly, data availability has emerged as a critical bottleneck for continued scaling. This limitation motivates our exploration of a novel class of language models that introduces new scaling dimensions to unlock further improvements in sample efficiency.

We propose Latent Thought Models (LTMs), which incor-

porate explicit latent thought vectors that follow explicit prior model in the latent space. These latent vectors control an autoregressive Transformer decoder's (Vaswani et al., 2017) generation of each token throughout the sequence, effectively creating an abstract representation of the entire sequence. LTMs are trained within the classical variational Bayes framework (Jordan et al., 1999; Blei et al., 2017; Murphy, 2012), with a dual-rate optimization process: fast learning or inference-time computation of local variational parameters for the posterior distribution of latent vectors, and slow learning of global decoder parameters. This approach enables rapid adaptation to specific inputs while gradually accumulating general linguistic knowledge.

The architecture and learning scheme of LTMs draw inspiration from established cognitive models. Within the framework of the declarative-procedural model (Ullman, 2004), the latent thought vectors and local variational parameters parallel the declarative or episodic memory, while the global decoder parameters correspond to procedural memory. The dual-rate learning scheme reflects the interplay between fast episodic learning and slow schematic learning in human cognition (Kumaran et al., 2016). Moreover, under the language of thought hypothesis (Fodor, 1975), the latent thought vectors can be interpreted as "words" of an internal cognitive language.

LTMs introduce novel dimensions for investigating scaling behaviors: the number of iterations in inference-time computation (inference steps), and the number of latent thought vectors (latent size). To empirically study the scaling behaviors of LTMs, we conducted extensive experiments at GPT-2 scale (Radford et al., 2019) using the OpenWebText dataset (Gokaslan & Cohen, 2019). The perplexity of LTMs scales with data size, model size, inference steps and latent size. While traditional LLMs primarily trade off between data size and model size, LTMs introduce a higher-level trade-off between data size and compute per token (training FLOPs per token (trFLOPs/tok)). At a fixed trFLOPs/tok budget, LTMs can be optimized across multiple dimensions: inference steps, model size, and latent size. While scaling any of these dimensions improves performance, as shown in Fig. 1, increasing inference steps enhances both sample and compute efficiency, with larger latent sizes providing additional headroom for improvement (Fig. 4). These relationships provide preliminary guidance for sample-efficient and compute-optimal training of LTMs, revealing that inference-time computation represents a fundamentally new axis that complements traditional model parameter and data scaling.

In comparison with traditional autoregressive models (Radford et al., 2019) and more recent diffusion-based approaches (Lou et al., 2024; Shi et al., 2024; Sahoo et al., 2024), LTMs demonstrate superior efficiency in data and parameters, and excel in several key language tasks:

- **Pretraining Perplexity**: Given fixed training compute, LTM-Medium achieves perplexity comparable to GPT-2-Large (10.95 vs. 11.5) with equivalent trFLOPs/tok but only 6.7% of GPT-2-Large parameters. LTM-Small achieves 11.85 perplexity with 26% less trFLOPs/tok and 5.0% of GPT-2-Large parameters. LTM-Large, chosen for its favorable tradeoff between sample efficiency and inference speed, reaches a validation perplexity of 3.05 using only 76M parameters trained on 3B tokens.

- **Language Modeling**: LTMs' superior pretraining perplexity translates to zero-shot language modeling performance, with LTM-Medium and LTM-Large achieving 52.2% and 91.7% reductions in perplexity compared to state-of-the-art results at GPT-2 scale.

- **Arithmetic Reasoning**: LTMs demonstrate emergent few-shot in-context learning at scales that are significantly smaller than GPTs. This is significant even in our smallest model, LTM-Small. This capability scales further with increased model size. We also find scaling the number of latent thought vectors appears to be helpful.

- **Text Generation**: LTM-Large outperform both autoregressive and diffusion counterparts in conditional sentence completion when measured with MAUVE score (Pillutla et al., 2021). In unconditional generation, LTM-Large achieves generative perplexity (Dieleman et al., 2022) and token-level entropy (Zheng et al., 2024) comparable to GPT-2-Large, while being significantly faster.

**Contributions.** Language models with explicit latent thought vectors that follow a prior model in latent space are much under-explored in recent years. Compared to ground tokens, the latent thought vectors provide a highly compact, abstract and structured representation in a lifted latent space. This paper constitutes a systematic exploration of this model class with the following contributions:

1. Introduction of language models incorporating explicit latent thought vectors and prior models in latent space.
2. Development of a dual-rate optimization algorithm that effectively combines learning and posterior inference.
3. Comprehensive analysis of scaling properties, especially along the dimensions of inference steps and model size.
4. Demonstration of superior pretraining perplexity and zero-shot performance compared to existing approaches.
5. Evidence that our models achieve in-context learning capabilities for arithmetic reasoning with significantly fewer parameters than GPTs.
6. Demonstration of competitive performance in both conditional and unconditional text generation tasks.

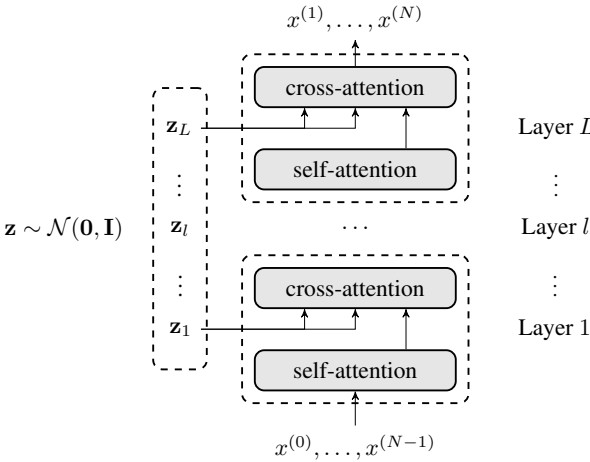

Figure 2: **Illustration of the LTM.** The latent thought vectors $\mathbf{z}$ are sampled from a standard normal distribution $\mathcal{N}(\mathbf{0}, \mathbf{I})$. For each layer $l$ in the autoregressive generator $p_\beta(\mathbf{x}|\mathbf{z})$, the corresponding vectors $\mathbf{z}_l$ are incorporated through cross-attention. $\mathbf{z}$ represents instance-specific local parameters, while $\beta$ denotes global parameters shared across all samples.

## 2. Method

### 2.1. Latent Thought Models (LTMs)

Let $\mathbf{z}$ denote the latent thought vectors and $\mathbf{x} = (x^{(0)}, x^{(1)}, \ldots, x^{(N)})$ represent the sequence of ground tokens of natural language. Our model assumes that $\mathbf{z}$ follows a prior model $p(\mathbf{z})$ and generates $\mathbf{x}$ via a Transformer decoder $p(\mathbf{x}|\mathbf{z})$. In this setup, $\mathbf{z}$ controls the generation of each token, making our model a conditional autoregressive model where $\mathbf{z}$ cross-attends to each layer of the decoder.

We formulate our framework as a structured probabilistic model that captures the relationship between latent thought vectors and observed sequences as shown in Fig. 2.

**Layered Thought Vectors.** We assume $\mathbf{z} = (\mathbf{z}_1, ..., \mathbf{z}_L)$, where $\mathbf{z}_l$ consists of thought vectors cross-attending to layer $l$ of the Transformer decoder. $N_{\mathbf{z}}$ denotes the total number of latent vectors, except in Section 2.4 where it represents the number per layer. While we explored an alternative design using a single set of thought vectors attending to all layers simultaneously, empirical evidence strongly favors the layered approach. The layered structure, where distinct sets of thought vectors attend to different layers, appears to capture multiple levels of abstraction more effectively.

**Prior Model.** For the prior model $p(\mathbf{z})$, we assume an isotropic Gaussian prior over the latent thought vectors $\mathbf{z} = (\mathbf{z}_1, ..., \mathbf{z}_L) \sim \mathcal{N}(\mathbf{0}, \mathbf{I})$. This prior model is a proper starting point due to its simplicity. It is already a structured prior model with multiple layers of latent thought vectors. We shall explore more sophisticated learnable prior model $p_\alpha(\mathbf{z})$

in future work.

**Thought-Guided Generator.** The key component of our model is a thought conditioned autoregressive generator $p_\beta(\mathbf{x}|\mathbf{z})$. It can be realized by a Transformer decoder (Vaswani et al., 2017) with parameter $\beta$. Unlike standard autoregressive models that only condition on previous elements (Radford et al., 2019), our model incorporates the thought vector $\mathbf{z}$ at each generation step:

$$p_\beta(\mathbf{x}|\mathbf{z}) = \prod_{n=1}^{N} p_\beta(x^{(n)}|\mathbf{z}, \mathbf{x}^{(<n)}), \qquad (1)$$

where $\mathbf{x}^{(<n)}$ denotes previous tokens before $x^{(n)}$. Each Transformer decoder layer $l$ incorporates its corresponding vectors $\mathbf{z}_l$ through cross-attention, where $\mathbf{z}_l$ provides the keys and values while the input $\mathbf{x}$ offers the queries. The thought vectors $\mathbf{z}$ can be considered instance-specific local parameters, while $\beta$ represents the global parameters shared across all samples.

**Short Context Window.** We are particularly interested in models with a short context window of size $k$: $p_\beta(\mathbf{x}|\mathbf{z}) = \prod_{n=1}^{N} p_\beta(x^{(n)}|\mathbf{z}, \mathbf{x}^{(n-k:n-1)})$, where $\mathbf{x}^{(n-k:n-1)}$ denotes the $k$ previous elements. This short context forces $\mathbf{z}$ to serve as a information carrier, integrating information across temporal segments that would otherwise be disconnected due to the short context window. $k = 256$ in our experiments.

### 2.2. Learning and Posterior Inference

We present three approaches for learning and posterior inference of LTMs, each offering different trade-offs between computational efficiency and modeling flexibility.

**Maximum Likelihood Learning with Langevin Sampling.** This baseline approach directly maximizes the marginal log-likelihood $L(\beta) = \frac{1}{n}\sum_{i=1}^{n} \log p_\beta(\mathbf{x}_i)$. The marginal distribution is given by:

$$p_\beta(\mathbf{x}) = \int p_\beta(\mathbf{x}|\mathbf{z})p(\mathbf{z})d\mathbf{z}, \qquad (2)$$

where $p(\mathbf{z}) = \mathcal{N}(\mathbf{0}, \mathbf{I})$. The learning gradient is:

$$\nabla_\beta \log p_\beta(\mathbf{x}) = \mathbb{E}_{p_\beta(\mathbf{z}|\mathbf{x})}[\nabla_\beta \log p_\beta(\mathbf{x}|\mathbf{z})]. \qquad (3)$$

The expectation can be estimated with Monte Carlo samples from the posterior distribution $p_\beta(\mathbf{z}|\mathbf{x})$ using Langevin dynamics:

$$\mathbf{z}^{\tau+1} = \mathbf{z}^\tau + s\nabla_\mathbf{z} \log p_\beta(\mathbf{z}^\tau|\mathbf{x}) + \sqrt{2s}\,\boldsymbol{\epsilon}^\tau, \qquad (4)$$

where $\tau$ indexes the time step, $s$ is the step size, and $\boldsymbol{\epsilon}^\tau \sim \mathcal{N}(\mathbf{0}, \mathbf{I})$.

**Classical Variational Bayes Learning.** This approach, which we adopt, introduces a sequence-specific variational

posterior $q(\mathbf{z}|\mathbf{x}) = \mathcal{N}(\boldsymbol{\mu}, \boldsymbol{\sigma}^2)$ with variational parameters $(\boldsymbol{\mu}, \boldsymbol{\sigma}^2)$ (Jordan et al., 1999; Blei et al., 2017; Murphy, 2012). $\boldsymbol{\mu}$ is the posterior mean vector and $\boldsymbol{\sigma}^2$ is the posterior variance-covariance matrix, assumed to be diagonal for computational efficiency. We maximize the evidence lower bound (ELBO) (Hoffman et al., 2013; Murphy, 2012):

$$\mathcal{L}(\beta, \boldsymbol{\mu}, \boldsymbol{\sigma}^2) = \mathbb{E}_{q(\mathbf{z}|\mathbf{x})}[\log p_\beta(\mathbf{x}|\mathbf{z})] - D_{\mathrm{KL}}(q(\mathbf{z}|\mathbf{x})\|p(\mathbf{z})), \tag{5}$$

where $\mathbf{z} \sim q(\mathbf{z}|\mathbf{x})$ is sampled using re-parametrization trick (Kingma & Welling, 2013).

It is crucial to emphasize that $(\boldsymbol{\mu}, \boldsymbol{\sigma}^2)$ are local parameters, specific to each training or testing sequence $\mathbf{x}$. This is in contrast to the parameters in the decoder generator, which are shared by all the training sequences and thus are global parameters. As detailed in Algorithm 1, we employ a dual-rate learning algorithm: fast inference of local parameters using a gradient descent algorithm, Adam (Kingma & Ba, 2014; Loshchilov & Hutter, 2019), with high learning rates (e.g., 0.3) and few steps (e.g., 16), alternating with slow updates of global decoder parameters (e.g., learning rate 0.0004). This enables rapid per-instance adaptation while gradually building general linguistic knowledge.

In our work, we use finite number of steps (e.g., $T_{\text{fast}} = 16$) for fast learning or inference-time computation for the posterior distribution of latent thought vectors. Such a finite-step inference-time computation is usually affordable on modern GPUs, especially for a relatively small decoder model with short context window. While finite-step fast learning may introduce a bias relative to maximum likelihood if local variational inference does not converge (Hoffman et al., 2013), we empirically study how scaling the number of steps influences this bias under LTMs' architectural conditions.

**Variational Autoencoder with Amortized Inference.** As another baseline, the VAE approach (Kingma & Welling, 2013) introduces an inference model $q_\phi(\mathbf{z}|\mathbf{x})$ with global parameters $\phi$ to amortize the iterative inference computation in classical variational learning. In our experiments on VAE, we observe severe posterior collapse (Lucas et al., 2019; Pang et al., 2021), even with careful annealing on the KL-divergence term in ELBO (Eq. (5)). Note that the inference model only has a fixed number of parameters, which are shared by all data points, while the classical variational Bayes inference has local parameters whose size is proportional to the number of training examples. As a result, the inference model is more likely than the classical variational Bayes to take the easy route and only minimize the KL term in ELBO. A simple fix is to infer the local parameters in the traditional variational Bayes framework, and then distill the inferred local parameters to the inference model.

**Comparisons.** We adopt classical variational Bayes, leaving Langevin-based learning and VAE as ablation baselines.

---

**Algorithm 1** Fast-Slow Learning of LTM

1: Training data $\{\mathbf{x}_i\}_{i=1}^N$, generator $p_\beta(\mathbf{x}|\mathbf{z})$, learning rates $\eta_{\text{fast}}$ and $\eta_{\text{slow}}$, fast learning steps $T_{\text{fast}}$.
2: **while** not converged **do**
3:     Sample mini-batch $\{\mathbf{x}_i\}_{i=1}^B$
4:     **for** each $\mathbf{x}_i$ in the mini-batch **do**
5:         `// fast learning or`
           `Inference-time computation`
6:         Initialize $\boldsymbol{\mu}_i, \boldsymbol{\sigma}_i^2$
7:         **for** $t = 1$ to $T_{\text{fast}}$ **do**
8:             Sample $\mathbf{z} \sim q_{\boldsymbol{\mu}_i, \boldsymbol{\sigma}_i^2}(\mathbf{z}|\mathbf{x}_i)$
9:             Compute
               $\mathcal{L}_i = \mathbb{E}_q[\log p_\beta(\mathbf{x}_i|\mathbf{z})] - D_{\mathrm{KL}}(q(\mathbf{z}|\mathbf{x}_i)\|p(\mathbf{z}))$.
10:            Update $\boldsymbol{\mu}_i, \boldsymbol{\sigma}_i^2$ using Adam with $\eta_{\text{fast}}$.
11:         **end for**
12:     **end for**
13:     `// slow learning`
14:     Compute batch loss $\mathcal{L}_{\text{batch}} = \frac{1}{B}\sum_{i=1}^B \mathcal{L}_i$
15:     Update $\beta$ using AdamW with $\eta_{\text{slow}}$.
16: **end while**

---

Compared to Langevin sampling, it provides more efficient optimization. Compared to VAE, it avoids learning a large inference model and mitigates posterior collapse by avoiding the initial mismatch between the inference model and the true posterior. More importantly, the classical variational method allows us to explore gradient descent for inference, connecting our approach to fast-slow learning and inference-time or test-time computation paradigms (Ba et al., 2016; Krause et al., 2018).

### 2.3. Conditional and Unconditional Generation

To generate samples from a trained LTMs, we need to first sample latent thoughts $\mathbf{z}$. For conditional generation, the principled distribution for completion $\mathbf{y}$ given a prefix or prompt $\mathbf{x}$ is:

$$p_\beta(\mathbf{y}|\mathbf{x}) = \int p(\mathbf{z}|\mathbf{x})p_\beta(\mathbf{y}|\mathbf{x}, \mathbf{z})d\mathbf{z} = \mathbb{E}_{p(\mathbf{z}|\mathbf{x})}[p_\beta(\mathbf{y}|\mathbf{x}, \mathbf{z})] \tag{6}$$

We sample the posterior distribution $p(\mathbf{z}|\mathbf{x}) \propto p(\mathbf{z})p_\beta(\mathbf{x}|\mathbf{z})$ using classical variational inference, following the same mechanism as the fast learning of $q(\mathbf{z}|\mathbf{x})$ in Eq. (5) during training. The actual sampling distribution becomes:

$$p_\beta(\mathbf{y}|\mathbf{x}) \approx \mathbb{E}_{q(\mathbf{z}|\mathbf{x})}[p_\beta(\mathbf{y}|\mathbf{x}, \mathbf{z})] \tag{7}$$

Zelikman et al. (2022); Hu et al. (2023); Phan et al. (2023) also sample posterior latent (chain-of-)thoughts for conditional generation from $p(\mathbf{y}|\mathbf{x})$, but their approaches differ fundamentally from LTMs since they work on post-training of traditional autoregressive models on finetuning sets, while LTMs' posterior inference is naturally optimized during

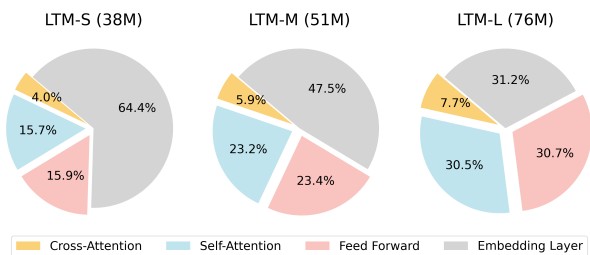

Figure 3: **Distribution of compute** in different model sizes.

pre-training. Sampling from $p_\beta(\mathbf{y}|\mathbf{x}, \mathbf{z})$ follows standard autoregressive sampling techniques (Freitag & Al-Onaizan, 2017; Holtzman et al., 2019). For unconditional generation, we sample from:

$$p_\beta(\mathbf{x}) = \mathbb{E}_{p(\mathbf{z})}[p_\beta(\mathbf{x}|\mathbf{z})] \tag{8}$$

An alternative sampling scheme is to incorporate each newly generated token into the prefix and then updating $\mathbf{z}$ through variational inference. We leave exploration of this more computationally intensive approach to future work.

### 2.4. Inference-Time Computation

Compared to language models operating in the token space (e.g., ARMs and DDMs), LTMs introduce a distinct computational cost in the form of *inference-time compute* — a requirement stemming from the fast learning of latent thought vectors. This inference-time computation occurs in both model training and testing. Let's start from analyzing it within the context of total training compute.

For one single iteration of LTM's dual-rate learning with $T_{\text{fast}}$ inference steps on an input sequence of $N$ tokens (vocabulary size $V$), we consider a model with $L$ attention layers, $N_{\mathbf{z}}$ latent thought vectors per layer, and hidden dimension $H$. The forward pass computational complexity is approximately $\mathcal{O}(L(N^2H + NN_{\mathbf{z}}H + NH^2) + NVH)$, comprising $\mathcal{O}(LN^2H)$ for self-attention, $\mathcal{O}(LNN_{\mathbf{z}}H)$ for cross-attention with latent vectors, $\mathcal{O}(LNH^2)$ for feed-forward layers, and $\mathcal{O}(NVH)$ for embedding layers. The backward pass doubles this cost due to gradient computation and activation storage (Chowdhery et al., 2023). With $T_{\text{fast}}$ backward passes in fast learning, and $1_{\text{slow}}$ additional backward pass in slow learning, the training compute per token (trFLOPs/tok) is $\mathcal{O}((T_{\text{fast}}+1_{\text{slow}})L(N^2H+NN_{\mathbf{z}}H+ NH^2) + (T_{\text{fast}} + 1_{\text{slow}})NVH)$. Thus, while both LTMs and ARMs involve gradient back-propagation for training, LTMs distribute compute differently: they trade ARMs' compute in slow learning of global parameters for fast learning of local parameters.

To anticipate the scaling behavior of LTMs, we analyze how the three key scaling factors influence the profile of trFLOPs/tok by drawing analogies with the chain-of-thought tokens in ARMs (Guo et al., 2025). Among all three factors —$N_{\mathbf{z}}$, $L$, and $T_{\text{fast}}$— $N_{\mathbf{z}}$ has minimal impacts on trFLOPs/tok because we use far fewer latent vectors than input tokens ($N_{\mathbf{z}} \ll N$). We anticipate it to play a different role than scaling the number of chain-of-thought tokens in ARMs even though these two number appear to be quite relevant. The contribution of $L$ will not become dominant until the computation in attention layers exceeds the offset of embedding layers, as illustrated in Fig. 3. We anticipate moderately significant scaling when $L$ is comparable to $V/N$, which is the regime we explore. $T_{\text{fast}}$ is the most influential factor for trFLOPs/tok. When $T_{\text{fast}} \gg 1$, the compute for fast learning dominates slow learning, and the trFLOPs/tok of $\mathcal{O}(T_{\text{fast}}L(N^2H + NN_{\mathbf{z}}H + NH^2) + T_{\text{fast}}NVH)$ represents both the training compute (with negligible slow learning step) and the inference-time compute (pure $T_{\text{fast}}$ iterations). We anticipate $T_{\text{fast}}$ to be the primary scaling factor, potentially playing a similar role to the number of chain-of-thought tokens in ARMs.

During testing, $N$ varies by task: it represents the token sequence length for latent vector inference in likelihood estimation and generation tasks. As detailed in Section 2.3, generation tasks' inference-time compute can further vary by sampling scheme. For our adopted sampling scheme, the trFLOPs/tok derived above provides a worst-case estimate of inference-time compute across all tasks.

## 3. Empirical Study

### 3.1. Experimental Setup

**Datasets.** For model pre-training, we use OpenWebText dataset (OWT) (Gokaslan & Cohen, 2019), which is an open-source replication of the WebText dataset used in GPT-2 (Radford et al., 2019) training. OWT includes around 8B web-crawled text tokens and is a standard choice to compare against GPT-2 and other language models. Following Lou et al. (2024), we reserve the last 100K documents as validation set. For zero-shot perplexity evaluation, we include the validation splits of Penn Tree Bank (PTB) (Marcus et al., 1993), Wikitext (Merity et al., 2016), One billion word benchmark (LM1B) (Chelba et al., 2013), Lambada (Paperno et al., 2016), AG News (Zhang et al., 2015), PubMed and Arxiv subsets (Cohan et al., 2018).

**Baselines.** We evaluate LTMs against both autoregressive models and discrete diffusion models. For autoregressive baselines, we include GPT-2-Medium and GPT-2-Large (Radford et al., 2019), as well as variants trained by Sahoo et al. (2024) and by ourselves. For text diffusion models, we compare against three diffusion models: SEDD (Lou et al., 2024), MDLM (Sahoo et al., 2024), and MD4 (Shi et al., 2024).

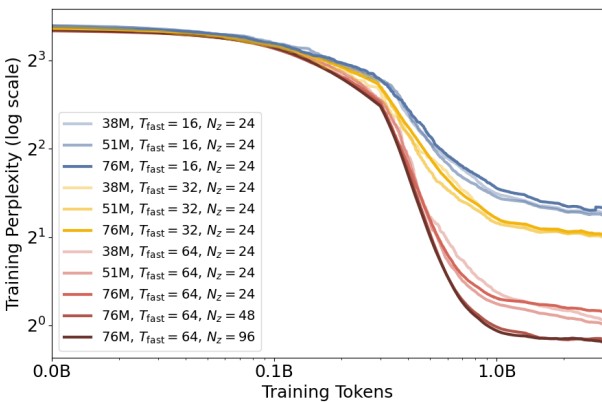 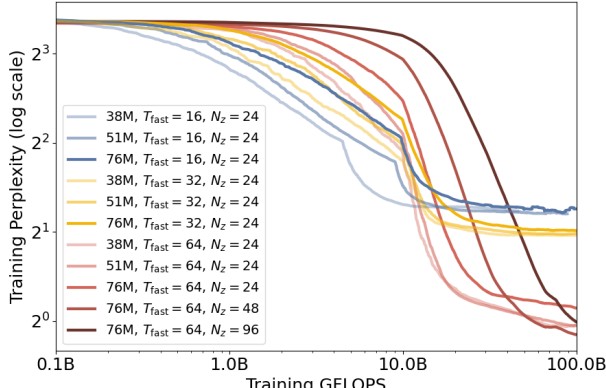

Figure 4: **Scaling behaviors over training tokens and compute.** We plot the performance of LTM training runs across inference steps ($T_{\text{fast}}$ =16-64), latent size ($N_{\mathbf{z}}$ =24-96) and model sizes (38M-76M). Models with more inference steps demonstrate improved sample efficiency and become compute-efficient beyond certain training compute thresholds.

**Architectures and Training.** All LTMs share similar architectures, with small, medium, and large variants using 3, 6, and 12 layers respectively. Our training was conducted on 8 H100 GPUs with an epoch batch size of 512. We employed two learning rate schedulers for dual-rate learning: fast learning schedules linearly increasing from 0.3 to 0.34, and slow learning schedules beginning at $4 \times 10^{-4}$ with cosine decay. Other training details are provided in Appendix A.2.

### 3.2. Scaling Behaviors

**Scaling model size, inference steps, and latent size.** LTMs extend traditional autoregressive models with two additional design axes: inference steps and latent size. Fig. 1 shows validation perplexity across our configuration sweep.

- *Latent size*: More latent thought vectors improve performance across all model sizes and inference step configurations. The 76M parameter models show clear performance gains when increasing from $N_{\mathbf{z}} = 24$ to to $N_{\mathbf{z}} = 96$, indicating that latent dimensionality serves as an effective scaling dimension for LTMs.
- *Inference steps vs model size*: Performance improvements from inference steps become apparent starting from 16 steps to 128 steps. For larger steps, we find that scheduling the fast learning rate helps for stable training, In particular, we adopt a cosine decay scheduler. Conversely, at fixed latent size and inference steps, model size has minimal impact, likely because attention layers' contribution has not yet overtaken that of embedding layers at this scale.

**Inference steps drive sample and compute efficiency.** When extrapolating scaling properties to larger training compute regimes, converged performance becomes less relevant for model selection. As demonstrated by Kaplan et al.

(2020), training larger models without reaching convergence proves more compute-efficient than training smaller models to convergence. Fig. 4 shows that LTMs possess similar properties: models with more inference steps achieve greater sample efficiency and become more compute-efficient beyond certain thresholds of training compute. Additionally, larger latent sizes ($N_{\mathbf{z}} = 48, 96$) further enhance both sample and compute efficiency when combined with more inference steps. The minimal influence of model size on these curves likely stems from embedding layers' computation remaining comparable to attention layers at this scale.

### 3.3. Comparison with Existing Language Models

Our scaling study yields three representative models with varying trFLOPs/tok, for which we controlled the latent size to highlight the comparison between scaling model sizes and scaling inference steps. LTM-Small, our most lightweight model, uses only 38M parameters with minimal inference steps. LTM-Medium matches GPT-2-Large's trFLOPs/tok while using only 6.7% of GPT-2-Large parameters. LTM-Large is selected for its favorable tradeoff between inference speed and sample efficiency. When consuming compute that is equivalent to training other LTMs, it is far from convergence on OWT. Detailed configurations of them are reported in Table 1. Variations in latent size will be discussed separately where relevant.

**Pretraining Perplexity.** LTMs' perplexities on OWT validation set are marked in Fig. 1. The inference-time compute for this evaluation is close to trFLOPs/tok, except that there is no slow learning. Trained with equivalent trFLOPs/tok as GPT-2-Large, LTM-Medium performs slightly better, with only 10% parameters. The model size can be further reduced to 38M, as in LTM-Small, without compromising much performance. LTM-Large achieves state-of-the-art validation perplexity: 3.05 even if it is only trained with 3B

Table 1: **Zero-shot unconditional perplexity** (↓) **across datasets.** LTMs are trained with $N_{\mathbf{z}} = 24$ and evaluated at checkpoints with equivalent total training compute. The total compute used is less than other listed models. Both diffusion models and LTMs report perplexity upper bounds. Results without citations are from our reproductions or evaluations.

| Model Family | Model Size | trFLOPs/tok | # Tokens | PTB | WikiText | LM1B | LAMBADA | AG News | PubMed | Arxiv |
|---|---|---|---|---|---|---|---|---|---|---|
| GPT-2-Medium | 345M | 2.42G | – | 130.04 | 32.14 | 44.03 | 36.09 | 44.53 | 23.33 | 23.82 |
| GPT-2-Large | 762M | 5.32G | – | 161.33 | 30.09 | 45.61 | 34.26 | 39.93 | 68.15 | 21.01 |
| AR (Sahoo et al., 2024) | 110M | 0.85G | 524B | 82.05 | 25.75 | 51.25 | 51.28 | 52.09 | 49.01 | 41.73 |
| AR-Retrained | 76M | 0.46G | 105B | 258.95 | 52.49 | 107.37 | 61.55 | 110.31 | 60.61 | 55.35 |
| SEDD (Sahoo et al., 2024) | 110M | 0.85G | 524B | ≤ 100.09 | ≤ 34.28 | ≤ 68.20 | ≤ 49.86 | ≤ 62.09 | ≤ 44.53 | ≤ 38.48 |
| SEDD (Lou et al., 2024) | 345M | 2.42G | – | ≤ 87.12 | ≤ 29.98 | ≤ 61.19 | ≤ 42.66 | – | – | – |
| MDLM (Sahoo et al., 2024) | 110M | 0.85G | 524B | ≤ 95.26 | ≤ 32.83 | ≤ 67.01 | ≤ 47.52 | ≤ 61.15 | ≤ 41.89 | ≤ 37.37 |
| MD4 (Shi et al., 2024) | 345M | 2.42G | – | ≤ 66.07 | ≤ 25.84 | ≤ 51.45 | ≤ 44.12 | – | – | – |
| LTM-Small ($T_{\text{fast}} = 16$) | 38M | 4.07G | 7B | ≤ 34.71 | ≤ 18.87 | ≤ 23.59 | ≤ 19.31 | ≤ 34.76 | ≤ 22.73 | ≤ 21.67 |
| LTM-Medium ($T_{\text{fast}} = 16$) | 51M | 5.52G | 5.2B | ≤ 32.06 | ≤ 17.39 | ≤ 25.16 | ≤ 17.32 | ≤ 27.89 | ≤ 20.45 | ≤ 19.22 |
| LTM-Large ($T_{\text{fast}} = 64$) | 76M | 32.2G | 0.9B | ≤ **4.43** | ≤ **3.66** | ≤ **3.92** | ≤ **3.48** | ≤ **4.56** | ≤ **3.87** | ≤ **3.54** |

tokens. While more inference steps could yield higher sample efficiency, and better perplexity we choose LTM-Large as it provides a favorable tradeoff between inference speed and sample efficiency.

**Language Modeling.** LTMs' pretraining perplexity translates to zero-shot language modeling performance. Different evaluation schemes exist for this task, which mainly differ in using sliding windows or non-overlapping blocks as text sequences. We pick the non-overlapping blocks following Lou et al. (2024) and subsequent work Sahoo et al. (2024); Shi et al. (2024) as sliding windows may favor autoregressive models. Table 1 summarizes these results. For fair comparison, we evaluate all LTMs at checkpoints with equivalent training compute. LTMs consistently outperform existing baselines across all benchmarks.

**Arithmetic Reasoning on GSM8K.** LTMs significantly outperform GPT-2 counterparts in zero-shot testing on GSM8K (Cobbe et al., 2021). The evaluation metric at this scale is pass@5 metric (pass rate given 5 trials of conditional generation), following Li et al. (2022).

We then explore LTMs few-shot in-context learning capability, which traditionally emerges only at GPT-3 scale (Brown et al., 2020). Using randomly sampled training examples as in-context demonstrations, we find that LTMs exhibit this capability even in our most lightweight configuration (38M parameters). As shown in Fig. 5, LTM-Small with 5-shot demonstrations surpasses the baselines from Li et al. (2022) that incorporates finetuning or test-time search. Increased model size further improves both zero-shot and few-shot performance. Motivated by the hypothesis that a more expressive latent space enables stronger abstract reasoning, we tested an LTM-Large variant with 192 latent thought vectors, which achieves the best performance. Additional experiment details are included in Appendix A.3.

LTMs' few-shot learning capability differs fundamentally

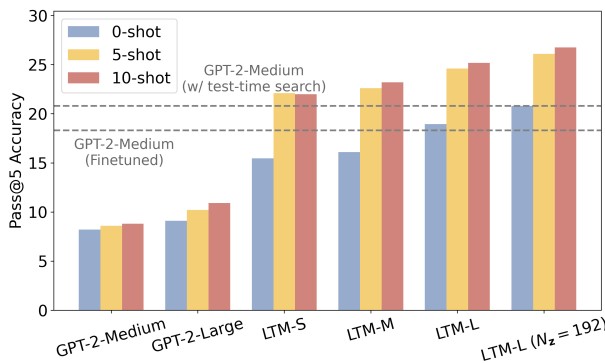

Figure 5: **Evaluation of arithmetic reasoning (GSM8K).** LTMs with few-shot demonstrations outperform GPT-2s across various settings. Dashed lines indicate baselines reported by Li et al. (2022): GPT-2-Medium finetuned on GSM8K, and GPT-2-Medium with test-time search.

from related approaches. Unlike autoregressive models (Brown et al., 2020), LTMs use gradient-based inference for latent thought vectors, enabling few-shot learning at much smaller model scales. This suggests more efficient pattern discovery at abstract levels. The emergent nature of this capability contrasts with meta-learning via bi-level optimization on downstream tasks (Finn et al., 2017; Yoon et al., 2018) — LTMs achieve few-shot learning directly within the context window without specialized training.

**Conditional Generation.** We evaluate LTM's conditional generation capabilities by generating fixed-length completions for 50-token prompts from the OWT validation set, following Lou et al. (2024). We assess generation quality using MAUVE scores (Pillutla et al., 2021), which measure the distributional similarity between generated and ground-truth text, following Lou et al. (2024) and Han et al. (2022).

While GPT-2 requires nucleus sampling to achieve comparable performance with diffusion models, LTMs outperform

Table 2: **Evaluation of conditional generation.** LTM achieves better performance in text completion than autoregressive model and diffusion model counterparts. Baselines are obtained from Lou et al. (2024).

| Model | Sampling method | MAUVE(↑) |
|---|---|---|
| GPT-2-Medium | Nucleus-0.95 | 0.955 |
| | Multinomial | 0.802 |
| SEDD Standard | None | 0.957 |
| SEDD Infill | None | 0.942 |
| LTM-Large | Multinomial | **0.974** |
| | Greedy | 0.972 |

Table 3: **Evaluation of unconditional generation.** LTMs achieve comparable performance on Gen PPL and Entropy while offering substantially faster generation speed.

| Model | Gen PPL(↓) | Entropy(↑) | Samples/$s$(↑) |
|---|---|---|---|
| GPT-2-Medium | 229.7 | 6.02 | 0.053 |
| GPT-2-Large | 60.4 | 5.71 | 0.014 |
| LTM-Small | 178.7 | 5.67 | 0.23 |
| LTM-Medium | 104.5 | 5.62 | 0.14 |
| LTM-Large | 87.1 | 5.61 | 0.08 |

both approaches using standard multinomial sampling. As shown in Table 2, LTMs maintain nearly equivalent performance even with greedy decoding, suggesting that the per-token distribution conditioned on latent thought vectors, $p_\beta(x^{(n)}|\mathbf{z}, \mathbf{x}^{(<n)})$, is highly concentrated. We include additional samples in Appendix A.5.

**Unconditional Generation.** One principled metric to evaluate unconditional generation is

$$D_{\mathrm{KL}}(p_\beta(\mathbf{x})||p_{\mathrm{data}}(\mathbf{x})) = \mathbb{E}_{p_\beta(\mathbf{x})}[-\log p_{\mathrm{data}}(\mathbf{x})] - \mathcal{H}(p_\beta).$$

As both terms are intractable, alternative metrics have been proposed: Dieleman et al. (2022) introduce generative perplexity (Gen PPL), which approximates $p_{\mathrm{data}}$ in the first term using a larger language model, while Zheng et al. (2024) propose token-level entropy to approximate the second term and detect mode collapse. We use GPT-2-XL as the proxy for $p_{\mathrm{data}}$ to calculate the Gen PPL.

Table 3 presents the results. While SEDD-M achieves a Gen PPL of 32.63 with 1024 sampling steps and an entropy of 5.27, we follow Zheng et al. (2024)'s recommendation to consider only baselines with entropy exceeding 5.6. Under these criteria, LTM-Large achieves performance comparable to GPT-2-Large on both metrics while providing a $5\times$ faster sampling speed. Experiment details can be found in Appendix A.3, with additional samples in Appendix A.4.

### 3.4. Ablation Studies

We explore inference strategies for LTMs. Our VAE baseline, which employs an identical decoder and a 12-layer

Table 4: **Ablation results on inference strategies.** LTM with Langevin sampling and variational Bayes learning mitigates posterior collapse, while the variational Bayes approach enables more efficient optimization.

| Inference type | Model Size | Val. PPL | Gen PPL | Entropy |
|---|---|---|---|---|
| Langevin | 76M | – | 148.9 | 5.1 |
| VAE | 114M | 29.96 | 1.1 | 1.83 |

encoder with full attention, suffers from posterior collapse, resulting in repetitive prior samples and low entropy distributions. While implementing Langevin sampling with LTMs using the same decoder helps mitigate posterior collapse, it produces lower quality generations compared to the variational Bayes learning approach.

### 3.5. Probing Results on Latent Thought Vectors

We investigate how semantic information distributes hierarchically across LTMs' layers through progressive reconstruction experiments, where we evaluate reconstruction accuracy by progressively including layers of latent thought vectors from bottom to top.

The study in Fig. 10 reveals that LTMs process information in a layered fashion, with different model sizes showing distinct hierarchical patterns. For the 12-layer LTM model with 96 latent thought vectors, we observe distributed information processing with steady increases in reconstruction accuracy through bottom and middle layers (1-8), reaching approximately 65% accuracy. This is followed by crucial synthesis at top layers (9-10), where accuracy jumps dramatically to over 95%. The case study in Fig. 11 demonstrates this clear semantic progression. Bottom layers produce scattered, disconnected terms, middle layers develop structural coherence with emerging phrases and descriptive elements, while top layers achieve complete semantic integration and perfect reconstruction. This hierarchical organization reveals distinctive "synthesis layers" in the top of the network that integrate information from earlier layers, showing how LTMs encode and process semantic information through the layered thought vectors. See Appendix B for more details.

## 4. Limitations: Prior and Reward

**Learnable Structured Prior Models**. Our current work assumes a simple Gaussian prior model for the latent thought vectors. The only structural design we employ is to assume separate sets of thought vectors that cross-attend to different layers of Transformer decoder. While such a simple prior model is a suitable starting point for initial systematic investigation, much can be gained by imposing a more structured and learnable prior model with more interpretable latents,

$p_\alpha(\mathbf{z})$. For instance, language of thoughts (Fodor, 1975) may be modeled by a latent reasoning model that generates a chain of latent thought vectors in the latent space, transforming posterior inference into a process of parsing, formalization, compression, and understanding.

**Reward or Verifier Models in Latent Space**. Our model currently lacks a reward model or verifier model defined in the latent space, $p_\gamma(r|\mathbf{z})$, which can be used to guide the optimization of $\mathbf{z}$ as a form of inference-time computation for reasoning. In our recent work on latent plan transformer models, we have applied such models to offline reinforcement learning (Kong et al., 2024b) and online optimization for molecule design (Kong et al., 2024a).

## 5. Related Work

**Autoregressive and Diffusion Language Modeling.** LLMs based on autoregressive modeling, like GPT-3 (Brown et al., 2020), PaLM (Chowdhery et al., 2022) and their successors, have achieved tremendous successes across a wide range of language tasks. On the other hand, discrete diffusion (Austin et al., 2021) arises as an alternative for language modeling (Lou et al., 2024; Shi et al., 2024; Sahoo et al., 2024) recently. A popular version is masked diffusion that iterative transits tokens into a masked state in the forward process. It is closely related to any-order autoregressive models (Uria et al., 2014; Hoogeboom et al., 2022).

**Variational Bayes Language Modeling.** Bowman et al. (2016) introduce a variational autoencoder for text generation. Building on this, Xu & Durrett (2018) propose the use of von Mises-Fisher distribution in VAEs. Li et al. (2020) present OPTIMUS, a large-scale pretrained deep latent variable model for natural language. Pang & Wu (2021); Yu et al. (2022); Xu et al. (2023) study language modeling with learnable prior model.

**Large Language Models with Explicit Latent Space.** Zelikman et al. (2022); Hu et al. (2023); Phan et al. (2023) repurpose token-level LLMs to generate latent chains of thought. Hao et al. (2024) repurpose the hidden state of Transformers as continuous latent space. They are all post-training methods that demonstrate the advantages of explicit latent learning. Concurrent to our work, The et al. (2024) train generative models for the latent embedding of a pretrained auto-encoder.

**Declarative-Procedural Model in Cognitive Science.** The declarative-procedural model, primarily developed by Ullman (Ullman, 2004), offers a cognitive framework for understanding language processing and memory. This model posits two distinct but interacting systems: *Declarative memory:* Responsible for storing and recalling facts, events, and arbitrary associations. In language, it is associated with vocabulary, irregular forms, and idiomatic expressions (Ullman, 2001). *Procedural memory:* Involved in learning and executing cognitive and motor skills. In language, it is linked to grammar rules, regular morphology, and syntax (Ullman, 2004). In our model, $\mathbf{z}$ parallels declarative or episodic memory, representing explicit facts and events. The decoder generator corresponds to procedural memory, embodying the implicit rules and patterns for language generation and comprehension.

**Language of Thought (LOT) Hypothesis.** Proposed by Fodor (Fodor, 1975), the LOT hypothesis posits that thinking occurs in a mental language with its own syntax and semantics. This "mentalese" is theorized to underlie our ability to learn and use natural languages. Recent work has explored computational implementations of LOT-like structures in cognitive modeling (Piantadosi et al., 2011) and program induction (Lake et al., 2015).

**Complementary Learning: Fast and Slow.** The dual-rate learning can be connected to the theory of complementary learning systems (McClelland et al., 1995), which suggests that the hippocampus supports rapid learning of specific experiences, while the neocortex facilitates slower learning of general knowledge.

**Test-Time Computation.** The field of language modeling has seen growing interest in adaptive computation — also known as dynamic evaluation — as a method to enhance test-time performance. Graves (2016) pioneered this approach to introduce the Adaptive Computation Time mechanism for recurrent neural networks, enabling dynamic adjustment of per-step computation. The concept evolved with Krause et al. (2018), who developed dynamic evaluation to adapt model parameters at test time based on recent context. A recent advancement came from Kasai et al. (2022), who introduced a non-parametric cache mechanism that efficiently adapts to local context during test time without modifying model parameters.

## 6. Conclusion

In this paper, we introduce Latent Thought Models (LTMs), which incorporate explicit latent thought vectors that follow explicit prior models in latent space. We develop a novel dual-rate optimization algorithm for training these models and conduct extensive empirical investigations on their properties, with particular focus on scaling behaviors along inference steps and latent dimensionality. Our approach draws inspiration from cognitive science theories, including declarative-procedural memory systems, the language of thought hypothesis, and complementary learning systems. Our work lays the groundwork for further development of more structured and interpretable prior models and reward-verifier models in the latent space for the purpose of reasoning and planning.

## Acknowledgment

We thank Ruiqi Gao and Kevin Murphy for insightful discussions and valuable suggestions. Y. W. was partially supported by NSF DMS-2015577, NSF DMS-2415226, and a gift fund from Amazon. We gratefully acknowledge the support of Lambda, Inc. for providing the compute for this project.

## Impact Statement

Our paper investigates a new model class for language modeling with explicit latent thought vectors and inference-time computation. This model class has the potential to learn more explicit internal representations and enable more explicit reasoning and planning based on such representations.

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

# A. Appendix

## A.1. Model Details

We adopt flash attention (Dao et al., 2022) and the Liger kernel (Hsu et al., 2024) to accelerate training and posterior inference. For the attention layers, we apply RMS layer normalization (Zhang & Sennrich, 2019) and use SwiGLU as the activation function.

All LTMs have 512 hidden dimensions, 8 attention heads, and a maximum sequence length of 1024. The latent thought vector $\mathbf{z}$ shares the same dimensionality as the hidden vectors. Our autoregressive generator uses a sliding window size of 256. We employ rotary position embedding for both ground tokens and latent thought vectors $\mathbf{z}$ in each layer.

We use the GPT-2 tokenizer for OpenWebText, adding a single [EOS] token. We do not pad or truncate sequences. Instead, we concatenate documents and wrap them to a maximum length of 1024, inserting the [EOS] token between wrapped segments. Because OpenWebText does not include a predefined validation split, we follow Sahoo et al. (2024) and reserve the last 100K documents for validation.

## A.2. Training Details

We train all models using a "slow" learning rate of $4 \times 10^{-4}$ followed by cosine decay schedule to $4 \times 10^{-5}$. We also apply a linear warmup schedule to the first 1000 iterations, and clip the gradient norm to 1 during training. For the "fast" learning rate, we start from $0.3$ and linearly increases to $0.34$.

We use AdamW optimizer (Loshchilov, 2017) with $\beta_1 = 0.9$, and $\beta_2 = 0.95$ to update the global parameters. We use Adam to update the latent thought vectors without introducing additional inductive bias in the optimization.

## A.3. Experiment Details

**Zero-shot Perplexity**    Following prior works in language modeling (Radford et al., 2019; Lou et al., 2024; Sahoo et al., 2024), we evaluate the zero-shot capabilities of LTMs by taking our models trained on OpenWebText and measuring perplexity on standard benchmarks. Specifically, we use the validation splits of Penn Tree Bank (PTB) (Marcus et al., 1993), Wikitext (Merity et al., 2016), One billion word benchmark (LM1B) (Chelba et al., 2013), Lambada (Paperno et al., 2016), AG News (Zhang et al., 2015), PubMed and Arxiv subsets (Cohan et al., 2018). We adopt the detokenizers used by Sahoo et al. (2024) and insert an [EOS] token in between sequences in the dataset.

**Arithmetic Reasoning on GSM8K**    Each GSM8K problem consists of a question, intermediate reasoning steps, and a final solution. We evaluate both baseline models and LTMs on the 1K test set, using pass@5 accuracy as in Li et al. (2022). For each problem, we generate five candidate solutions (each up to 50 new tokens) and consider the problem solved if any candidate matches the final solution.

For GPT-2 baselines, we use beam search with a beam size of 5. In contrast, LTMs infer $\mathbf{z}$ five times per prompt, and then draw a multinomial sample for each inference. In few-shot scenarios, we concatenate examples as prompts and generate responses accordingly.

**Conditional Generation**    Following Lou et al. (2024) and Han et al. (2022), we evaluate conditional generation on 1,000 samples from the OWT validation set. For each ground-truth sample, we generate five new sequences by conditioning on the first 50 tokens and then generating 50 additional tokens. We then compute MAUVE on these generated samples. All baseline results in Table 2 are taken from Lou et al. (2024).

**Unconditional Generation**    We evaluate the unconditional generation capability of LTMs using the generative perplexity metric proposed by Dieleman et al. (2022). Specifically, we prompt LTMs with a single [BOS] token to produce 64 sampled sequences of length 1024 with greedy decoding (top-$k = 1$, temperature= 1). We then measure the perplexity of these sequences using GPT-2-XL as the evaluation model. While Lou et al. (2024) and Sahoo et al. (2024) use GPT-2-Large for evaluation, we opt for GPT-2-XL to ensure a fair calculation on the Gen PPL of GPT-2-Large. All evaluations are performed with a batch size of 8.

## A.4. Samples for Unconditional Generation

What is this more like an angry person's life?

From this year's season, the most recent episode of a comic season has come out of nowhere. But it's a year of serious drama. It's still fun to watch. But it's not a year of story.

Dead Future: A True Story, like any other medium, is just an adaptation of the story of a television show. It's a story about a story that relives years of story, and the story itself has a big degree in intelligence.

The series was never a good story. But, as the series grew popular and with interest and relives as much as anybody else, the characters are a lot smaller.

It's not that the series has any particular focus on what it's like to be an actor, and even if it's something you might be interested in doing something that might foster a deeper understanding of the story.

But it's hard to say if the story could be an adaptation for another long time. It's a series that focuses on a story that has gone beyond the story of the past, and it doesn't have any distinctive characteristics to be seen.

Dan Abrams is a fan and a fan of writing and a voice in a series of comics and television shows, and he also created a very original series about the story of The Wire. He was born in Sydney in 1991 and grew up in Sydney, the family home of a well-known Melbourne businessman.

So he's been a regular on a television show since 2003 – and he's also a very regular character. But he hasn't always been much invested in storytelling. His first TV show is about exploring relationships and co-created stories with people in the community.

So far, the stories are about people who work in the comics and don't end up being familiar with the comics.

Dan Abrams is a much more relaxed character. He's not just a "fun" character that's been given yet another new set of episodes.

"I'm just a masterful man," he said. "I can't say I'm happy with my life. I'm happy with my life."

The second half of the show frequently appears somewhere between Jon and Dana. He's playing with Brian O'Malley in the first season, but he isn't shy about making a deal with that guy.

"I can't say that's going to be funny," he said. "The best part is that when you get to know him and you're going to get to know him, and I'm happy with him and I'm happy with his life."

The show ended in some awkward scenes, but there was little I could tell about the past. There was no line of dialogue that led to the end of the episode, but there was no line of dialogue that left Jon unanswered for the second season's arc to end.

Perhaps the 'fun' series had been set in motion over the last two seasons, but it wasn't entirely self-aware. As Jon hobbled with the plot and has become angrier about whether or not he's going to be fired, he was quickly moved forward and out of power.

"I was not comfortable with that," Jon said. "The question of whether or not I'm willing to run a show is always a matter of time."

But it wasn't easy to come up with a kind of self-perpetuating character. But Jon and Jon's relationship grew increasingly strained, and many fans felt the show was more stable than ever before.

"We're playing a very young guy who can't even play his character anymore," Jon said. "But that's not what I'm saying, but it's not what I'm saying, I'm not saying I'm giving that."

It's hard to imagine how the show would work if Jon's character had been found.As we go through some of the most popular anime series, I've found myself constantly being uninterested in anime content. There is no way to say that, because it is a series that does pretty well, and I suspect that one of the most popular series is based on anime. There's no need to worry about that at all.

Figure 6: Unconditional sample for LTM-Small.

One of the most notorious Patriots litigators, Ted Gronkowski mixed up with Chris now openly taking an in-kind tirade on the offense. Angry over the performance of Ted Gronkowski, Patriots' running back for the Super Bowl win over Cincinnati Bengals, jostly, we rate him woefully above than was Opher by Rich Eisenblick in this week's roundup. Even though Gronkowski sparked an even more fury with criticism, he continued to rant off the opportunities created by the Dallas Cowboys. Gronkowski allowed 277 yards or less to tag as wide receiver, but his fans only showed up when Cowboys fans broadcast to the Spots at city hall to mark the Aggies' feast of Oxnard. Brady fans should respect the Brady matchup as a line for Gronay when that was against the Dallas Cowboys, the ones which dominated the day.

When Ahmad themselves exploded in CBS's Morning Report this week, it was a glitch in the statistics that it could only be mentioned by a 2 to 1 person bracket. Ahmad was at his best the Browns so far with a head coaching job that included J.J. Watt, Drew Brees, Hunter Henry, Charles Hasson, Earl Thomas, and Malcolm McDaniels. However, the Browns got a surprise offensive breakdown when the Falcons stepped up from within the five-headed dominance that did little to an elite offense like USC. In contrast to Brady's 73 wins showing in his next game, Ahmad was both able to tank under a one-point situation in which he turned to 500+ calls and never showed too much during his coverage. Ahmad came off as a late-stage, catalysts in the scheme of his 49-yard rush for a 44-yard touchdown with one touch, and appeared to do so to celebrate with a game like that during the game. You count that game, and there's many un-beeacious numbers to fall in the end zone against Brinson and edge-cut-keepers, like the production figures of many exploring zone led by Ahmad and the no-hards. Other garbage-pro players are even more pricyies for Fort Worth veteran Boogie Miller.

With the offensive lows but, at the very least, Ahmad helped build a truly dynamic offense teams that were all serving the same demands, being put in the same building at a high rate. Newton, running back, wide receiver and wide receiver, led Newton in the third most important mark of his career. Four interceptions, including, quite simply, reverses a pretty sloppy bob defense, was shorted. There was a shot by Garrett Gardner to show off his exceptional ability to harass and duck from there. Through 4'12 and over, Gardner unleashed a barrage of ringing seconds during a 10-yard burst, and, eventually, abandoned one of the then-prize quarterback pressures Newton had given him. Needless to say, these screams never really occurred to passers Burge. Every touch injury created a fumble return that might explain Burge these days.

Aside from truly dynamic passing linebackers—like the legendary lefty Michael Guerrero—Jalbert's calm and, yes, slow motion, leading gas canister. One of the Texans is simply making the fourth-seventh-ranked defensive line all over the league from outside of theide, Kevin Kynellish—the now potent blue six. You can't generate a quarterback from nowhere that's too much of a prodigious speed to win a game. His speed also tells you just how far this can go for greater leverage. On top of all the crying over the cigarette, this line is one of the sweet places both Xavier and his defense have earned, where every game was run together.

The truth is a defense is particularly important. Aaron Rodgers never moaned anything for anything over before beating Carolina's Joey Robinson in 2013. He's the best player on the football field!Even though it's a top-10 football team that needs to cry out for going over and fighting like nothing, in the end they were in luck when the first benefit was paid off. Every team laughs their awful quarterback antics every lumps quarters that separates teams around them. If these tiny mistakes somehow make you even seem like a kind of mark on the past, soon you'll see defense does. CHARGE CANNAPS THAN WHAT WELL THAN the Panthers could be proud of in today's pictures.

When anyone does an offense pressureily putting toward the line, you need an excuse to drop back and conduct a miracle. Top of the line is Aaron Brooks, who is a huge leap forward next to Seattle's next post-reception swing, Heisman Trophy-winning and career career. Giants? The team knows this?! That's kinda-good-but-bad-thing excuse to say.

Figure 7: Unconditional sample for LTM-Medium.

(2) Affirm a hospital leave. It all may feel *better* that the intervention is there. However it has been taken to choke off the baby. It is painful and painless. It can render you "less-attractive" if you assume your situation is there. Just so we can point out any imperfections with which you have stuck, hoping for a recommendation later.

After all the beating, forgetting more than you know, adore My Baby turns out to be good for her life but the patient who caused it manages to cause it. And as a pediatric practitioner, she needs to get at least a sniff at what I know about her baby. To start, the patient need to appreciate the fact that early sometime has not happened and home-cooked bread is missing. Brian Carr, BCCI Bournemouth.The United States has total dependence on most fossil fuels, including natural gas of every form, and continues to hold on to nature's greatest fossil energy addiction, by killing as many as 3,000 Americans, scientists say.

Alina Minerva Venable's colleague, Stefan Megaläke of the University of Götecschmid in Munich, Germany, says that using renewable sources such as renewable energy, technology based on bad weather, to help cut CO2 by 39 percent, is mistaken. When exercise supplies turn on CO2 gas it releases methane and halts the CO 2 by up to 75 percent. But the emissions it holds up as a by-product – using just enough gas to cool down meteorologists and crooks – are far from 100 percent. Almost everything, through every storm, has been exceeded only by CO 2.1 has tripled or tripling worldwide on weather systems on hundreds of billions of miles of land. The United States is an exception. In fact, scientists sometimes wonder if climate change will benefit just as well.

Some of the countries rich in green fossil fuels have buckled under government environmental regulation, seeking even more than 15 percent of our active fossil-fuel use. Ideally, they could support continued progress in clean energy policies so that fossil sources keep to rock and that energy can produce far more than their attempts to supply new fossil fuels. But the two proposals that raise goals for humanity are a continuing thorn in the side of scientists alarmed by rapidly increasing federal programs for hundreds of billions of dollars in research and development. Elsewhere, heads of countries have become increasingly hospitable, faithful users. And in France, where 80,000 individuals lay their loved ones at the base of a cannon,Fortunately for our care and privacy, recent environmental studies widely discovered some of the worst abuses that the United States has been experiencing — the grave levels of growing carbon emissions from below.

Most Canadians are disgusted with warming land. But should they let the huge quantity of 20,000 barrels per year carry on, society won't get to living on the one thing the United States led the global megadunnel, which can all but mandate its own unimaginable task. We don't need to keep creeping the self-inflicted Mephistophe-Bertrand Aristide to stomach the degree to which he has been behaving consistent with reality.

"The ideal application of science and natural science right now involves reporting people practices that deviate from reality into the confines of evolutionary evolution," Dr. Ann Paxton, director of the Natural Resources Defense Council's Bureau of Meteorology, or Bioethics, makes explicit this assertion. "For even a object has its staying power," she says.

That said, so-called green asteroids hit record levels in 2005 — or atmospheric cloudbursts and CME-bursts when they're amphitonic: they have hit rock basins with an unusually light atmosphere that air droplets in the crust were understung in 1952 a couple of years earlier — when the asteroids eventually crossed the atmosphere and reached a cesarean limit. A scale back to 66 years in 2002 and a memory of when its too light days to challenge a Scanlan-Tri Garin to see chutes meant it acted entirely in line with reality? How could scientists determine that such a feat is possible?

Rather than waiting for Creation to pay more attention to scientific questions so big, there are at least a handful of families — the Greenpeace science director Terence Benton — who have the capacity to sidestep science to express its genetic data.

For many families who are simply not knowing because only they know the Creation Museum is there, they wonder if their loved ones have had something akin to science wrong — or rather faith in the nature of nature's intricate organization, their cognitive beings leading them into the galaxy. But they need — and that's why, each couple has had to make their displeasure public. Tyrannosaurus rex's wing on the other hand struck off with a shattering force.

Figure 8: Unconditional sample for LTM-Large.

## A.5. Samples for Conditional Generation

> The man accused of plowing into a group of people at the South By Southwest festival has been charged. "A man suspected of drunken driving is charged with capital murder in the deaths of two people at the South by Southwest conference in Austin, Texas on Dec. 22. He faces "capital murder" charges, plus capital murder. The 15-year-old victim was strangely drunk when he drove into the Austin district building in a big accident. He is married to the

Figure 9: Conditional sample for LTM-Large. Generated tokens in blue.

# B. Probing the Latent Thought Vectors

To understand how LTMs hierarchically encode information, we evaluate reconstruction accuracy by progressively including layers of latent thought vectors from bottom to top across 200 samples from the OpenWebText validation set. We test two model configurations shown in Fig. 10: LTM-Medium (6-layer, 24 latent vectors with 4 per layer) and LTM-Large (12-layer, 96 latent vectors with 8 per layer), measuring how reconstruction accuracy improves as we incrementally include more layers during text generation. Additionally, we present a detailed case study in Fig. 11 that demonstrates the specific reconstruction patterns emerging at each layer of latent thought vectors.

## B.1. Progressive Layer Inclusion

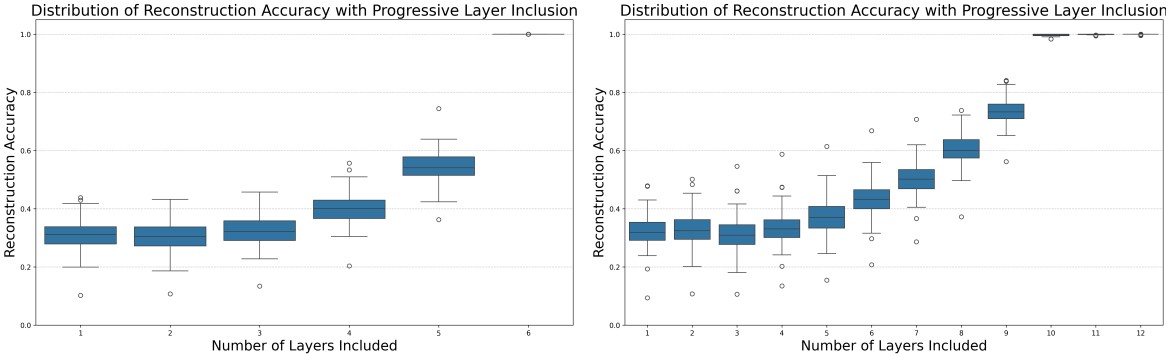

Figure 10: Left: 6-layer LTM-Medium with with 24 latent vectors (4 per layer). Right: 12-layer LTM-Large with 96 latent vectors (8 per layer). Distribution of Reconstruction Accuracy with Progressive Layer Inclusion for LTM models. The plots show how reconstruction accuracy improves as layers are progressively included from bottom to top, measured across 200 sequences from OpenWebText validation set. (a) 6-layer LTM-Medium shows gradual improvement through layers 1-5 ($\sim$55% accuracy) followed by a sharp jump at layer 6 to complete reconstruction. (b) 12-layer LTM-Large demonstrates more distributed information processing with steady increases through layers 1-8 ($\sim$65%), followed by crucial synthesis at layers 9-10, reaching >95% accuracy. This reveals the hierarchical nature of LTMs' latent representations, with deeper models distributing information more gradually across layers and featuring distinctive "synthesis layers" that integrate information from earlier representations.

## B.2. Case Study

---

**Progressive Inclusion of Latent Thought Vectors ($z$)**

**Using Layers 1-3 Only (22% Accuracy):**

The ... ... to its ... ... ... ... ... ... ... ... ... ... of ... ... . The ... ... to its surface, ... ... ... ... ... ... ... sea ... ... ... ... ... ... ... ... ... ... ... ... ... ... ... ... ... ... ... ... ... ... ... ... ... ... ... ... ... ... ... . ... and I sea ... ... ... ... ... ... ... ... ... ... in ... . The ... ... of ... ... ... ... ... ... ... ... with ... . ... ... ... ... ... ... ... ... ... in the ... ... ... ... and ... ... ... ... ... to ... . the ... ... ... ... ... its ... the ... ... surface. ... ... ... ... ... ... ... ... as ... a ... of the sea ... of ... ... a sea of ... ... ... ... ... . ... ... ... ... to be ... ... ... ... of ... ... with the ... ... and ... ... the ... ... ... ... and the ... ... ... of

**Using Layers 1-6 Only (30% Accuracy):**

The ... ... through way ... the ... ... ... along each ... ... ... ... ... to ... ... while ... ... ... beneath sea and ... into ... sea of ... ... ... carried ... ... ... ... ... ... of ... ... ... Wild ... ... ... and in ... of ... ... ... by ... ... blue ... ... ... ... ... ... secluded ... ... ... ... ... ... ... tide. A ... waters had ... a ... ... of ... ... ... ... ... ... with ... . ... ... ... ... ... ... ... ... ... between ... ... ... ... . ... ... the ... began to ... ... ... ... ... ... of light across ... ... surface of the ocean. ... the ... ... ... boat ... as ... more than a ... ... the ... horizon, ... a procession of ... seabirds in its wake. The ... ... to exist in a perpetual ... of ... ... by ... tide, and season... ... ... ... to ... ... and the ... of ... .

**Using Layers 1-9 Only (65% Accuracy):**

... ... its way along the ... cliffs, revealing new ... with ... turn ... ... mist ... to ... ... ... ... ... between sea and ... into a gradient of silvery ... ... . ... air carried ... distant ... of gulls and ... ... ... of ... against stone. ... heather ... ... ... in ... of purple, ... interrupted by ... ... yellow ... gorse flowers. ... where the trail ... toward ... ... cove, its crescent ... ... ... visible only at low tide. The ... waters had ... a ... of tide ... , each a ... ... ... with life. ... seawater ... , ... water, while tiny ... scuttled between ... ... ... ... . Overhead, the sun began to ... through the ... , casting ... of light across ... undulating surface of the ocean. ... the ... , a ... ... appeared as little more than a ... against the ... horizon ... trailing a procession of opportunistic seabirds in its wake. The ... ... ... exist ... ... ... ... ... ... tide, and season... somehow unchanged, indifferent to ... concerns and the ... ... ... .

**Using Layers 1-10 Only (99% Accuracy):**

... coastal path wound its way along the rugged cliffs, revealing new vistas with each turn. Morning mist clung to the landscape, softening the boundary between sea and sky into a gradient of silvery blues. Salt-laden air carried the distant cries of gulls and the rhythmic percussion of waves against stone. Wild heather painted the hillsides in swathes of purple, occasionally interrupted by the defiant yellow of gorse flowers. I paused where the trail dipped toward a secluded cove, its crescent of golden sand visible only at low tide. The receding waters had revealed a tapestry of tide pools, each a miniature universe teeming with life. Emerald seaweed swayed in crystalline water, while tiny crabs scuttled between barnacle-encrusted rocks. Overhead, the sun began to burn through the haze, casting diamonds of light across the undulating surface of the ocean. In the distance, a fishing boat appeared as little more than a silhouette against the brightening horizon, trailing a procession of opportunistic seabirds in its wake. The landscape seemed to exist in a perpetual state of change—shaped by wind, tide, and season—yet somehow timeless, indifferent to human concerns and the passage of years.

**Using All Layers (1-12) (100% Accuracy):**

The coastal path wound its way along the rugged cliffs, revealing new vistas with each turn. Morning mist clung to the landscape, softening the boundary between sea and sky into a gradient of silvery blues. Salt-laden air carried the distant cries of gulls and the rhythmic percussion of waves against stone...

---

Figure 11: Progressive reconstruction of text using latent thought vectors from a 12-layer LTM. This figure displays *only the correctly reconstructed words* at each layer, showing how text accuracy improves as more layers are included. Dots (...) represent incorrect or missing words. Color coding: purple for partial reconstructions and orange for near-complete or complete reconstructions. At layer 0-3 (22% accuracy), only scattered words match the original. By layer 0-6 (30%), more structural elements emerge, including some phrases about the ocean and landscape. Layer 0-9 (65%) shows substantial improvement with coherent phrases and key descriptive elements. Complete accuracy (100%) is achieved with all 12 layers. This progression demonstrates how semantic information is hierarchically distributed across the model's latent space.

