# OpenReview forum: "Latent Thought Models with Variational Bayes Inference-Time Computation"
_ICML.cc/2025/Conference — ICML 2025 poster_

### Official Review · Reviewer_ReTi · 2025-03-09

**Overall Recommendation:** 3

**Summary:**

This paper presents a novel method called latent-thought language model (VTM) for autoregressive language modeling.
VTM introduces an additional family of scaling dimensions, latent thought vectors, to implicitly learn a sequence representation and guide the generation.
Training VTM requires sampling from the variational posterior of the latent vector, which is done by MCMC or variational Bayes.
The resulting method enjoys a good scaling property by achieving a better perplexity score or sample accuracy with a smaller number of tokens and model sizes.

**Claims And Evidence:**

The claims of this paper are supported by clear and convincing evidence.

**Essential References Not Discussed:**

No

**Experimental Designs Or Analyses:**

- It is not sure that the model indeed learns informative latent vectors. The authors could demonstrate the posterior distribution of $z$ given a sequence $x$; For example, $z$s may have a large correlation if $x$s have the similar semantic meaning.
- No computational cost in terms of training times or memory are reported.

**Methods And Evaluation Criteria:**

There are some potential issues with the implementation of the method.
- The authors incorporate a latent variable $z$ in each layer of the transformer, as illustrated by Figure 2. This makes the mask attention in classical autoregressive models not applicable, and will require $L$ NFEs to evaluate the probability.
- An additional variational posterior model $q_{\theta}(z|x)$ is employed to train the VTM model, which may introduce at least 2x memory. By the way, the authors do not explicitly explain how they design $q_{\theta}(z|x)$.
- The authors state that they use cross-attention to incorporate $z$, which is still vague for me. I suggest the authors use mathematical expressions or illustrations to explain how the generative model and the variaitonal posterior model are designed.

**Other Comments Or Suggestions:**

I have no other comments.

**Other Strengths And Weaknesses:**

I have no other comments.

**Questions For Authors:**

I have no other comments.

**Relation To Broader Scientific Literature:**

I have no other comments.

**Theoretical Claims:**

There are no the theoretical claims in this paper.

---

> ### Author Rebuttal · Authors · 2025-04-01
>
> *Thank you for your thoughtful review acknowledging the novelty of our work. We appreciate your recognition that our claims are supported by clear and convincing evidence. We will address your concerns as follows.*
>
> **1. Clarification on the cross-attention mechanism.**
>
> We wish to humbly clarify a potential misunderstanding regarding the reviewer's comment that incorporating $z$ makes "mask attention in classical autoregressive models not applicable." Our Latent Thought Models (LTMs) adopt the decoder Transformer in the original transformer paper of (Vaswani et al. 2017.), where each layer $l$ of the decoder transformer consists of (1) self-attention sub-layer with attention of $x_t$ to $x_{\leq t}$ (masked), (2) cross-attention sub-layer with attention of $x_t$ to $z_l$, and (3) feed-forward sub-layer. The cross-attention sub-layer (2) for each layer $l$ is ${\rm softmax}\left(\frac{Q_{h_l}K_{z_l}^\top} {\sqrt{d}}\right)V_{z_l},$ where $h_l$ is the hidden embedding of the input $x$ and $z_l$ consists of the latent thought vectors at layer $l$. Crucially, the latent vectors $z$ are inferred by gradient-based variational inference. This means there is no network that maps from $x$ to $z$.
>
> It might be a slight misunderstanding that LTMs "will require $L$ NFEs to evaluate the probability", if $L$ means the number of layers. Given a latent $z$, evaluating $p(x|z)$ only needs 1 NFE. The reviewer correctly notes additional computation is needed, but this is for approximating the posterior $p(z∣x)$ or evaluating the ELBO. This variational inference requires $T_{fast}$ NFEs, scaling with the inference steps $T_{fast}$, not layers $L$.
>
> **2. Memory requirements of additional variational posterior model (possibly 2x)?**
>
> We wish to respectfully clarify that the 2x memory requirement regarding our additional variational model may be a slight misunderstanding. The classical variational Bayes of our LTMs does not require a separate "variational encoder model" - it only needs the local parameters $(\mu, \sigma)$. The memory overhead for these parameters is almost negligible, with our LTM-L incurring just 1.5% additional memory cost on H100 compared to same-sized AR models with the same batch size. This is more efficient than the encoder (inference model) used in amortized variational inference (VAEs), which typically requires the same memory as the decoder.
>
> **3. Demonstration that model learns informative latent vectors (e.g., posterior distribution analysis)**
>
> Following your advice, we conduct in-depth analysis of our latent thought vectors through progressive latent inclusion experiments as [probing results (link)](https://anonymous.4open.science/r/icml25_submission3398_probing-F2BB/). Our findings reveal: (1) Clear hierarchical latent representation in LTMs, with higher layers integrating information from lower ones. LTM-M shows a significant accuracy jump at the final layer (55% to 100%), while LTM-L demonstrates more gradual information distribution with a notable improvement at layers 9-10 (65% to 95%). (2) Qualitative text reconstruction reveals progression from basic grammatical elements (22% at layers 1-3) to structural patterns (30% at layers 1-6) to coherent semantic content (65% at layers 1-9) to complete sentence (99% at layers 1-10).
>
>
> **4. Computational cost reporting (training time, memory usage)**
>
> We reported the trainFLOPs/tok and the number of training tokens in Table 1 of our submission. Here we provide other costs such as training time and memory usage.
>
> | Model | Parameters | Memory Usage | Time per Epoch | Gradient Accumulation |
> |-------|------------|--------------|----------------|------------------------|
> | GPT-2-L | 762M | 52214 MiB | 924.63 ms | 4 |
> | GPT-2-M | 345M | 62706 MiB | 453.71 ms | 2 |
> | LTM-M (steps=16) | 51M | 59040 MiB | 2209.76 ms | 1 |
> | LTM-L (steps=64) | 76M | 68158 MiB | 13190.58 ms | 1 |
>
> All benchmarks were conducted on a single node with 8 H100 GPUs with batch size 512. In our approach, we trade compute for slow learning of global parameters for fast learning of local parameters (line 242-246), which is essential to LTM's data efficiency.
>
> *We will revise our paper based on your valuable feedbacks. Thank you!*

---

### Official Review · Reviewer_YgR1 · 2025-03-11

**Overall Recommendation:** 2

**Summary:**

This paper proposes a probabilistic language models called Latent-Thought Language Models (LTMs), which introduce explicit latent vectors to layers of transformers.  The authors claim that this setup yields new “scaling dimensions” beyond traditional LLMs, allowing more efficient use of training compute per token. They report gains in both sample efficiency and parameter efficiency, as well as improved validation perplexity, zero-shot language modeling, and conditional/unconditional text generation.

**Claims And Evidence:**

- Supported Claims:

The paper’s experiments compare LTMs with standard autoregressive models and discrete diffusion models, showing quantifiable improvements in perplexity and sample efficiency—these results appear well-supported by the reported metrics.
- Potentially Under‐Supported Claims:

Latent Thought Vectors as a “Language of Thought”
The cognitive analogy is intriguing but not strongly backed by empirical studies.
Multiple Inference Steps

The statement that “increasing inference steps” yields better performance seems reasonable, but the exact algorithm (and how it differs from typical amortized VI) is not fully explained. Why is it necessary to infer for multiple steps?

**Essential References Not Discussed:**

No.

**Experimental Designs Or Analyses:**

- Experimental Soundness:
The overall design—comparing against standard LLMs and a discrete diffusion baseline—makes sense. Reporting perplexities and a few-shot test is appropriate.
- Potential Weakness:
The paper does not sufficiently detail hyperparameter choices or the exact inference procedure (especially the “multiple inference steps”), potentially leaving some designs unclear.
The authors claim that latent vectors store global context, but there is no direct analysis (like an attention map or latent dimension probe) confirming that they indeed capture such information.

**Methods And Evaluation Criteria:**

- Methods:

The authors’ methodology of adding latent vectors to a Transformer decoder, learned via a variational approach, makes sense.

Evaluation Criteria:

Validation Perplexity and zero-shot performance are standard metrics for assessing language models, so these are appropriate.
Additional benchmarks (e.g., text generation tasks, in-context reasoning tasks) improve the assessment.

**Other Comments Or Suggestions:**

N.A.

**Other Strengths And Weaknesses:**

-- Key Strengths:

- empirical gains in perplexity and few-shot reasoning.
- Offers new “scaling dimensions” and suggests paths for more efficient training or inference.

-- Key Weaknesses:

- Vague details about the practical inference procedure and reproducibility.
- Some conceptual leaps (e.g., short windows forcing global information into latent variables) without robust empirical backing; the relation between the method design of this work and cognitive concepts )

**Questions For Authors:**

See above

**Relation To Broader Scientific Literature:**

Original research seems to incorporate VI with pre-trained models for fine-tuning, while this work investigates training starting from scratch. The significance of this work becomes the engineering part. However, the code and implementation are not provided.

**Theoretical Claims:**

N.A.

---

> ### Author Rebuttal · Authors · 2025-04-01
>
> *Thank you for your constructive feedbacks. Below are our responses.*
>
> **1. Comparison to amortized VI and clarification on multi-step inference**
>
> We wish to humbly clarify a possible misunderstanding in terms of the Variational Inference (VI) in our Latent Thought Models (LTMs). LTMs employ the classical variational Bayes (VB) framework rather than amortized inference (as in variational autoencoders (VAEs)).
>
> As detailed in Sec. 2.2 and Algorithm 1 (line 165-183), for each training sequence $x$, the posterior $p(z|x)$ is approximated by a variational distribution $N(\mu, \sigma^2)$. Note that $(\mu, \sigma^2)$ are local parameters specific to each sequence $x$, and are optimized by maximizing the evidence lower bound (ELBO) for each $x$. To clarify the reviewer's comment about "multi-step inference": it refers to the iterative update of $(\mu, \sigma^2)$, where the number of steps is the number of gradient ascent steps for maximizing the ELBO for each $x$.
>
> In contrast to our VB approach, in amortized inference all the training sequences share an encoder (or inference model) with global parameters $\phi$. The encoder maps each $x$ to the corresponding $(\mu, \sigma^2)$ directly. There is no iterative update in typical amortized encoders.
>
> Our ablation study in Sec.3.4 confirmed that classical VB achieves better ELBO and avoids VAE's potential posterior collapse problem.
>
> **2. Implementation and hyperparameters**
>
> We described our inference algorithm on page 4, Algorithm 1. We discussed the key hyperparameters (`inference steps` and `latent size`) in Sec. 3.2. Other hyperparameters can be found in Appx. 1 & 2. To further address your concern, we have built anonymous links for both [pseudocode (link)](https://anonymous.4open.science/r/icml25_submission3398-F84E/) and [executable code (link)](https://anonymous.4open.science/r/icml25_submission3398_code-4F9D).
>
> **3. About novelty of our work regarding variational inference**
>
> To our knowledge, our work on incorporating classical VB in language models is novel, as confirmed by Reviewer FqJd and Reviewer ReTi. We are unaware of prior work using classical VI for fine-tuning LLMs. If we are provided with any related references, we would be happy to cite them in revision. The explicit latent abstraction in our model can be an important direction to explore for language modeling.
>
> **4. Empirical backing of cognitive analogy of "Language of Thought"**
>
> Thanks for your comment. Our model was conceptually inspired by the "Language of Thought" model in cognitive science. However, there has not yet been a machine learning model for it. Our work may be considered the first step towards such a model. To further address your concern, we conduct in-depth analysis of our latent thought vectors through progressive latent inclusion experiments as [probing results (link)](https://anonymous.4open.science/r/icml25_submission3398_probing-F2BB/), where our findings reveal:(1) Clear hierarchical latent representation in LTMs, with higher layers integrating information from lower ones. LTM-M shows a significant accuracy jump at the final layer(55% to 100%), while LTM-L demonstrates more gradual information distribution with a notable improvement at layers 9-10(65% to 95%). (2)Case study reveals that lower layers capture grammatical elements and higher layers encode increasingly complex semantic content.
>
> **5. Direct analysis confirming latent vectors capture global information**
>
> In lines 149-164, the latent vectors $z$ controls the generation of each token in $p(x_{t}|z, x_{<t})$ via cross attention, and $z$ is inferred from the whole sequence $x$: $p(z|x) \propto p(z) \prod_t p_\beta(x_t|z, x_{<t})$, where $\prod_t$ is over all $t$.
>
> Empirically, we validate this by reconstruction experiments where we first infer $z$ from the sequence $x$ and then generate the sequence using only these inferred $z$ (without conditioning on ground truth context). This yields 100% reconstruction accuracy on the OpenWebText validation set, confirming that $z$ successfully captures the global information necessary for generation.
>
> This finding is further supported by our probing results across different models (link is mentioned by response to point 4).
>
> **6. Empirical support for conceptual claims (short windows forcing global information into latent variables)**
>
> In our response to your point 5 above, $z$ controls the generation of each $x_t$, and is inferred from the whole sequence $x$. With the short context window, the generation of $x_t$ must rely on $z$ to provide information beyond the short window. Empirically, our inferred $z$ can reconstruct the whole sequence accurately as mentioned above. Our probing results (link is mentioned by reponse to point 4) further confirm that the inferred $z$ captures the global information.
>
> *We will improve our paper based on your helpful feedbacks. Given our responses above, we humbly request you to reconsider your rating. Thanks!*

---

### Official Review · Reviewer_FqJd · 2025-03-14

**Overall Recommendation:** 4

**Summary:**

This work presents novel Latent-Thought Language (LTM) class of models, where the explicit latent vector is introduced to guide the generation of tokens. The model is optimized withing the variational bayes framework, using faster learning rate for latent vector distribution parameters and slower rate for learning decoder parameters. Empirical evaluation against autoregressive and diffusion models highlighted few results: increased parameter efficiency, improved validation perplexity, and showing the few shots in-context reasoning capability.

## update after rebuttal
After reading other reviews and author responses, I decided to keep my initial grade.

**Claims And Evidence:**

The claims of increased parameter efficiency, improved validation perplexity, and few shots in-context reasoning capabilities are in general well supported by empirical evidence.

**Essential References Not Discussed:**

Other flavors of Large language Models like Mixture of Experts Models as well as State Space Models were not considered or discussed.
Jiang, Albert Q., Alexandre Sablayrolles, Antoine Roux, Arthur Mensch, Blanche Savary, Chris Bamford, Devendra Singh Chaplot et al. "Mixtral of experts."
Gu, Albert, and Tri Dao. "Mamba: Linear-time sequence modeling with selective state spaces."

**Experimental Designs Or Analyses:**

Experiment design and analysis appear sound. Ablation is performed. Appropriate metrics like perplexity and MAUVE scores are used.

**Methods And Evaluation Criteria:**

Yes, the proposed methods and evaluation criteria are appropriate for the problem. Well know benchmark sets are used: OpenWebText for pretraining,  seven open datasets for zero shot unconditional tasks and GSM8k for few shots evaluation of arithmetic reasoning.

**Other Comments Or Suggestions:**

Typos:
Page 2, right column, line 109: "is not as good as have separate sets"

**Other Strengths And Weaknesses:**

Strong points:
* The idea seems quite novel
* Strong performance against GPT2, SEDD, MDLM and MD4.

Weak points
* Scalability might be difficult
* While overall clear, the paper is "dense" at some moments. And certain phrases require more explanation, for example what is referred by "lifted latent space"

**Questions For Authors:**

1. How expensive is the inference, and how does the inference latency compares to other LLM approaches?
2. Were all compared models using the same GPT2 tokenizer?

**Relation To Broader Scientific Literature:**

Main idea is inspired by declarative procedural model (Ullman, M.T. Contributions of memory circuits to language: The declarative/procedural model. Cognition,92(1-2):
 231–270,2004.) and language of though (Fodor, J.A. The Language of Thought. Harvard University Press,1975.). Most technical details are related to variational Bayes literature.

**Theoretical Claims:**

N/A

---

> ### Author Rebuttal · Authors · 2025-04-01
>
> *Thank you for your insightful comments and for recognizing the novelty and strong performance of our work. We address your concerns point-by-point as follows.*
>
> **1. Scalability.**
>
> Our Latent Thought Models (LTMs) scale along two primary dimensions: model size and the number of inference steps ($T_{fast}$). This parallels scaling model size and scaling the length of CoTs in Auto-Regressive (AR) models. Crucially, as discussed in Sec. 2.5, the FLOPs/token cost from $T_{fast}$​ scales linearly, similar to CoT length cost. Therefore, LTMs pose no fundamental asymptotic barriers to scaling beyond those of current LLMs.
>
> In practice, we do acknowledge the engineering challenge of building robust infrastructure for large-scale gradient-based Variational Inference (for pre-training and deployment) and designate this as future work.
>
> **2. Inference expense and latency compare to other LLMs**
>
> Our analysis in Sec. 2.5 indicates that the inference expense (FLOPs/token) scales linearly with the number of inference steps, $T_{fast}$​. To offer a concrete comparison: performing 16 inference steps with our LTM-M model has latency comparable to generating ~8.7 CoT tokens in the iso-trainFLOPs/tok AR baseline, GPT2-L.
>
> It is also necessary to highlight that practical inference expenses vary significantly depending on the task. While likelihood estimation (Table 1) processes the full sequence length, conditional generation (Fig 6 & Table 2) benefits from typically shorter prefill contexts. Furthermore, for unconditional generation (Table 3), there is no gradient backpropagation in LTMs at inference time, which may lead to considerable speedups compared to other LLM baselines whose decoder networks are significantly larger.
>
> **3. Were all compared models using the same GPT2 tokenizer?**
>
> Yes. (line 612 in Appendix)
>
> **4. Explanation of "lifted latent space"**
>
> Thank you for the feedback. "Lifted" is synonymous to "latent" in our context, which is meant to emphasize abstraction from "ground" tokens. We shall make this clear in revision.
>
> **5. Typo on page 2: "is not as good as have separate sets"**
>
> Thanks. We will fix it in revision.
>
> **6. Discussion of other language models like Mixture of Expert(MoE), State Space Model(Mamba) etc.**
>
> We will cite these LLM architectures in revision and discuss building our LTMs based on these architectures in future work. Our cross-attention layers that incorporate $z$ can be naturally inserted into these architectures. Since at each cross attention layer, $z_l$ is compact (consisting of only a small number of vectors), the cross-attention is very light, and this is actually quite consistent with the attention-less state vector in Mamba. We shall explore this direction in future work.
>
> *We will revise our paper based on your deeply insightful comments. Thank you for your positive evaluation!*

---

### Decision · Program_Chairs · 2025-05-01

**Decision:**

Accept (poster)

**Comment:**

I recommend accepting this paper on Latent-Thought Language Models. This novel approach incorporates explicit latent thought vectors that guide token generation through a Transformer decoder, utilizing classical variational Bayes for optimization. The key strengths include superior sample and parameter efficiency, demonstrated improvements in perplexity and few-shot reasoning capabilities. This work offers a promising new direction for language modeling with compelling scalability properties along multiple dimensions. I think the authors did a good job addressing the reviewers' concern including the one who gave weak reject rating.